# Mitochondrial Genome Variations and Possible Adaptive Implications in Some Tephritid Flies (Diptera, Tephritidae)

**DOI:** 10.3390/ijms26125560

**Published:** 2025-06-10

**Authors:** Natalia S. Medina, Manuela Moreno-Carmona, Nelson A. Canal, Carlos F. Prada-Quiroga

**Affiliations:** 1Maestría en Ciencias Biológicas, Facultad de Ciencias, Universidad del Tolima, Ibagué 700062, Tolima, Colombia; nsmedinac@ut.edu.co (N.S.M.); manuelaamoreno@ut.edu.co (M.M.-C.); 2Grupo de Investigación de Biología y Ecología de Artrópodos, Facultad de Ciencias, Universidad del Tolima, Ibagué 7300062, Tolima, Colombia; 3Grupo de Investigación en Moscas de las Frutas, Facultad de Ingeniería Agronómica, Universidad del Tolima, Ibagué 7300062, Tolima, Colombia; nacanal@ut.edu.co

**Keywords:** mitochondrial genomes, comparative genomics, Tephritidae adaptations, bioinformatic analyses

## Abstract

Tephritidae is an economically important family among Diptera that also exhibits high diversity, biogeographical distribution, and different lifestyles. Despite the recent release of genomes and mitochondrial genome sequences of various species of the family, the evolutionary history of the group and the origin of host adaptation within it remain poorly understood. We undertook a whole-mitochondrial-genome study covering molecular variation at the mitochondrial level by analyzing 10 new mitochondrial genomes obtained from genomic data reported and downloaded from the SRA database from NCBI, analyzed in FastQC and assembled through MITGARD, and 44 mitogenomes available in the Organelle—Refseq database, in total representing 4 subfamilies, 9 tribes, 13 genera, and 54 species. We determined compositional asymmetry and codon usage patterns across the different subfamilies analyzed by using DNASp6 and CAICal. We found high evolutionary rates in the NADH genes, which could play an important role in the adaptation of species to different hosts and environmental variation. By using maximum likelihood phylogenetic reconstruction obtained by IQTREE and ModelFinder, and lifestyle and distribution data of the included species, we considered a generalist feature, explained as possible predominant adaptation in some members of the family. This study in Tephritidae tries to demonstrate possible patterns among molecular variability in mitogenomes, adaptations, and lifestyles. Our findings suggest that selection pressures on certain NADH genes may be linked to host specificity in some Tephritidae species, providing evolutionary insights into how molecular evolution drives ecological adaptation or biogeographical diversity, probably in response to changing environmental conditions and host–parasite co-evolution across taxa.

## 1. Introduction

Tephritidae are acalyptrate flies with a wide global distribution (except in extreme desert and polar areas), with more than 5000 species [1,2]. Some species in the family are parasitoids or saprophages; however, the vast majority are phytophagous. The family includes a variety of agriculturally important pests that attack a wide range of fruits and fleshy vegetables, posing significant food security challenges worldwide [3]. These flies exhibit a variety of complex and interesting behaviors, and some have been used as models in evolutionary or ecological studies [4]. Numerous basic and applied studies have focused on the pest species of this group [5,6], but there are also many on non-pest, largely non-numerous frugivorous tephritid species [5,6,7].

Phylogenetic analyses of the family include those by Korneyev [8], on morphological characters, and by Han and Ro [9] and Han and McPheron [10], using molecular markers; however, our understanding of the relationships within the family is unsatisfactory [9]. There are 6 subfamilies and 27 tribes [6,8,9]. The only biology known for Tachiniscinae is that one species is a parasitoid; Blepharoneurinae develop in Cucurbitaceae; Phytalmiinae are mainly saprophagous; Trypetinae exhibit a diverse range of feeding behaviors, with a few presenting predatory feeding on different plant tissues and Dacinae being found in the stems of Poaceae or fruits or seeds of a variety of plant families (Liliaceae); and Tephritinae are specialized in the infestation of *Asparagus* (Zaceratini) or Asteraceae and a few other families [6,11]. Knowledge of the biology and ecology of Tephritidae and adaptative mechanisms of host use could help to understand the evolution of the family.

Morphological and molecular analyses have been performed to understand the evolutionary history within this family, highlighting specific sexual traits of females for oviposition in hosts [7,12,13,14], individual size [7], or sensory receptive capacity of organisms to recognize different fruits for their development [14,15,16]. Likewise, molecular markers such as microsatellites [17], nuclear ribosomal genes, genes with specific functions [18,19], or mitochondrial genes [20] allow for a deeper understanding of the most important subfamilies and genera of Tephritidae to gain further knowledge about their relationships and habits. However, there are still unexplored areas, especially in the molecular field, which are currently being analyzed to continuously contribute to the study of this family.

The insect mitochondrial genome (mitogenome) has been widely used as a source of data for phylogenetic inference and evolutionary studies due to reduced gene content, rapid evolution, maternal inheritance, and its role within the cell, which includes the production of most ATP, the control of the cell cycle, and cell growth [21,22,23,24]. Due to their crucial cellular function, a change in metabolic demand may result in selection pressure on mitochondrial protein-encoded genes (as well as in other non-coding genes, such as tRNAs and rRNAs) to satisfy the energy production processes, specifically in the electron transport chain, which can have a significant impact on the metabolism of the organism’s lifestyle, possibly in conjunction with nuclear genes; so, adaptive selection is suggested to be a major influence on mitochondrial evolution [25,26,27]. In this context, a variety of analyses of strand asymmetry and codon usage bias (CUB) of mitochondrial genomes in species of insects have been related to evolution, suggesting that higher selection pressure, such as gene length, gene function, and translational selection, dominates the codon preference of mtDNA, while the composition constraints for mutation bias only play a minor role [28,29,30]. Therefore, CUB can be considered an excellent tool to study the evolution of animal mitochondrial genomes due to the role it plays in important cellular processes such as transcription, mRNA stability, translation efficiency, and accuracy [31,32,33].

Comprehensive mitochondrial genome data sets [34], as well as coding and ribosomal genes from the mitogenome, have been used for phylogenetic reconstruction in Tephritidae, primarily for economically important tephritid genera such as *Bactrocera* [35], *Ceratitis* [36], *Dacus* [37], and *Anastrepha* [38]. At the mitogenome level, research has focused on the analysis of protein-coding genes to study neutrality and selective pressures [39]. These studies have suggested a possible relation between molecular variability and adaptative forces generated by the metabolic requirements of lifestyle in different organisms, as well as the influence of extrinsic factors, such as altitude, temperature, availability of resources, hosts, and habitat [40,41]. These results suggest that adaptative selection plays an important role in mitochondrial evolution, although specific analysis in Tephritidae has been limited to only one genus [41].

New sequencing technologies and bioinformatic methodologies have made possible multiple processes to obtain new mitogenomes through the assembly of genomic or transcriptomic information to obtain complete and reliable mitogenome sequences and low percentages of error rates [42,43,44,45]. However, to date, only 47 complete mitogenome sequences of Tephritidae are accessible in GenBank (https://www.ncbi.nlm.nih.gov/, accessed on 16 September 2024), which is less than 3% of the more than 5000 reported species; thus, publication of new mitochondrial genomes is needed to discover the molecular characteristics of the family more fully. The focus of this study was to obtain the mitogenomes of ten additional species of Tephritidae and to compare the mitogenomes of all Tephritidae available to date. We also tried to associate the variation in mitogenomes with selection processes, adaptations, or lifestyles of the species via an analysis not only at the phylogenetic but also adaptive level in the family. We hypothesized that molecular variability in the mitogenome, including nucleotide composition, codon skew, and codon usage, as well as the influence of selective or mutational pressure on mitochondrial genes, could be associated with the different lifestyles of Tephritidae species. We expected high metabolic plasticity in generalist individuals compared with specialist organisms restricted to specific host plants.

## 2. Results and Discussion

### 2.1. Assembled Mitogenomes and Features

We assembled 10 mitochondrial genomes of the following Tephritidae subfamilies, tribes, and species: *Dacus axanus* (Hering), 1938 (Dacinae, Dacini); *Rhagoletis batava* (Hering), 1958; *R. pomonella* (Walsh), 1867; *R. zephyria* (Snow), 1894; *R. completa* (Cresson), 1929; *Carpomya incompleta* (Becker), 1903; *C. vesuviana* (Costa), 1854 (Trypetinae, Carpomyini); *Euleia heraclei* (Linnaeus), 1758 (Trypetinae, Trypetini); *Tephritis californica* (Doane), 1899 (Tephritinae, Tephritini); and *Merzomyia westermanni* (Meigen), 1826 (Tephritinae, Eutretini). Previous species belong to genera that are unrepresented in the NCBI RefSeq database due to the limited availability of mitogenome sequences. We obtained genomic reads ranging from 1,546,528 to 230,764,584 of total reads, allowing us to assemble complete mitogenome sequences. From these data, we assembled mitochondrial reads ranging from 20,113 to 182,047 bp in length and extracted mitogenome sequences with lengths ranging from 16,362 bp in *R. zephyria* to 15,056 bp in *T. californica*. NCBI accessions and other information such as average read depth are summarized in Appendix A.

All 10 mitogenomes assembled contained the typical 13 protein-coding genes (PCGs), two ribosomal RNA genes (*rrnS* and *rrnL*), and 22 transfer RNA (tRNA) genes, as well as the non-coding region called control region (CR) (Appendix A). Our results indicate that all of these tephitid mitogenomes show the same synteny, as has been proposed for mitogenomes of Insecta [23] and Hexapoda [46]. This has been previously reported for *Bactrocera* and *Dacus* [47,48,49,50,51,52,53] and confirms that these tephritid species are characterized by a conserved gene order in their mitochondrial genome [23,46].

### 2.2. AT/GC Content and Skew Analysis

We performed a nucleotide content analysis in our 10 newly obtained mitogenomes and the 44 downloaded mitogenome sequences. For all sequences, the average AT% was 73.94 ± 2.92, with variation among the subfamilies. For example, the species of Dacinae have a lower AT concentration (73.0% ± 2.21) than Trypetinae (75.5% ± 2.72) and Tephritinae (79.7% ± 0.99); the nucleotide composition ratio of A > T > C > G in all 54 sequences is shown in Appendix A. These molecular features are consistent with previous reports for Tephritidae due to the presence of non-coding regions throughout the mitogenome [35,49,53] and variations in the percentage of A + T nucleotide content in PCGs [50,51]. Ambiguities could be the result of sequencing problems to obtain reads used for the assembly process [54] or incomplete references to obtain the mitogenome molecule; also, the presence of hypervariable regions in genomes increases the possibility of finding these ambiguities in different levels or percentages [55].

We performed an AT/GC skew nucleotide content analysis at different levels (whole-genome, PCGs, and tRNAs levels). At the whole-mitochondrial-genome level, we found a bias towards adenine in the AT skew (0.0659 ± 0.0253) and a bias towards cytosine in the GC skew (−0.2265 ± 0.0452). However, GC skew variation among subfamilies was observed. While, in Dacinae, the average in the GC skew was 0.0730 ± 0.025, in Trypetinae, it was 0.0457 ± 0.010 and in Tephritinae 0.0486 ± 0.013 (Figure 1A and Appendix A). These values are similar to previous analyses of *Bactrocera* [34,35,49,50,51], *Dacus* [48], *Felderimyia* [56], *Zeugodacus* [57], and *Ceratitis* [58]. In other Diptera, there are reports of the same pattern at the infraorder level in Tipulomorpha, Muscomorpha, Culicomorpha, Tabanomorpha, and Asilomorpha [59]. Other orders of Insecta, such as Coleoptera, Hymenoptera, Lepidoptera, Neuroptera, and Megaloptera, also have positive AT and negative GC skew values [60], meaning that this skewed distribution of nucleotides is likely influenced by selection processes and asymmetrical mutation patterns during replication and transcription [29,61], generating complementary sequences with increased A and C contents [29,62,63].

At the PCG level, species within subfamilies showed varying biases in the GC and AT skew, with principal differences in Trypetinae, which had a bias towards guanine for all species in the GC skew (0.0249 ± 0.0146). In Tephritinae, half of the species showed a bias towards guanine (0.0226 ± 0.0006) and the other half to cytosine (−0.0027 ± 0.0026), with lower values in the GC skew, while Phytalmiinae and Dacinae had shared features (Figure 1B and Appendix A). A thymine bias in the AT skew was observed in all species (−0.1519 ± 0.0076), which has been found in other families of Diptera, such as Anisopodidae, Anthomyiidae, Keroplatidae, and Syrphidae [59]. The negative skewness towards thymine in AT has not been deeply analyzed in Insecta, although in species such as *Aleurodicus dugesii*, *Neomaskellia andropogonis*, and *Aleurochiton aceris* (Hemiptera) and in the subfamily Oedipodinae (Orthoptera), the positive skew bias towards guanine in AT/GC could be explained as a possible strand asymmetry reversal in the heavy strand of the mitogenome [29].

Individual PCG comparisons revealed conserved features within genes, such as a negative AT skew with a bias towards thymine (−0.0608 ± 0.1664) and a negative GC skew towards cytosine (−0.2363 ± 0.1184) in most of the genes. Likewise, the *nad1*, *nad4*, *nad4l*, and *nad5* genes have a shared highly positive AT skew with a bias towards adenine (0.2842 ± 0.1787) and an average negative GC skew with a bias towards cytosine (−0.3204 ± 0.0727) (Figure 1C and Appendix A). A negative skew towards thymine in AT in PCGs, as we obtained in our results, has also been found and analyzed in other members of Tephritidae, such as *Sphaeniscus atilius* [64], and other Diptera species, such as Anthomyiidae [59] and *Blepharipa* sp. and other Oestroidea [65]. This could be linked to the third-codon positions in genes with this feature, indicating positive skews in AT in the light strand and negative or positive skews in AT in the heavy strand [29]. This character could be linked to genome replication orientation or direction in gene position, as has been found in Phthiraptera, Hemiptera, and Hymenoptera [29]. An important variability in nucleotide composition has been found in certain PCGs (*atp8*, *nd4l*, *nad6*, and *cox2*), in Dacinae species (*Bactrocera* and *Dacus*), detecting outliers or atypical data in these four genes in this group of species (Figure 1C and Appendix A).

The tRNA gene analysis showed similar distribution patterns and high dispersion in the data, with a positive AT skew towards adenine (0.0274 ± 0.0148) and a negative GC skew towards cytosine (−0.1092 ± 0.0283) (Figure 1D and Appendix A), following the general pattern in other Tephritidae, such as *Bactrocera* [34,35,49,50,51], *Dacus* [48], *Felderimyia* [56], *Zeugodacus* [57], *Ceratitis* [58], and *Sphaeniscus atilius* [64]. In other Diptera, the same pattern is shared in the AT skew in some species of Culicidae (*Haemagogus*) [66] and Sepsidae (*Nemopoda mamaevi*) [67].

The analysis indicates a negative correlation between AT skew and GC skew values across the entire mitochondrial genome (R = −0.89; Figure 1A), similar to what was observed when analyzing all PCGs together (R = −0.52; Figure 1B) or individual PCGs (R = −0.46; Figure 1C) and for each tRNA gene (R = −0.79; Figure 1D). The analyzed sequences in this and subsequent analyses show that species of Tephritidae and Trypetinae differ from Dacinae, specifically species of *Bactrocera*, due to the presence of higher AT content and lower AT/GC skew asymmetry in the latter (Appendix A). In this sense, previous studies postulate that the mtDNA replication process might be considered the major source of GC skew variation in animal mitogenomes [68]. Thus, this possible asymmetric mutation process in genes of great metabolic importance could be used to analyze phylogenetic relationships and evolutionary adaptations in species of Tephritidae as in other insect species.

### 2.3. Relative Synonymous Codon Usage (RSCU) and Codon Adaptation Index (CAI) Analysis

This is the first comprehensive study of relative synonymous codon usage (RSCU) in the mitochondrial genome of Tephritidae using 13 PCGs. These data show a high bias in certain codons, where the most used codon is TTA (Leu), followed by TCT (Ser), CGA (Arg), and TCA (Ser). Likewise, the codons with the lowest codon usage were AGG (Ser) followed by CTG and CTC (Leu) (Figure 2 and Appendix A).

In general, subtle differences in codon usage among different species in a family were found, such as synonymous variation in TTA (Leu), TTT (Phe), and AAA (Lys). For example, the RSCU of TTA is 3.47 among species of Dacini and Acanthonevrini, compared with 4.87 in Ceratitidini, Toxotrypanini, Trypetinae, and Tephritinae, with minimum values of 2.056 in *Bactrocera tsuneonis* (Dacinae) to a maximum of 5.29 in *Tephritis californica* (Tephritinae). These results could indicate a possible association between codon bias and phylogenetic relationship, where a higher bias in the use of these codons is observed in groups such as Trypetinae and especially in Tephritinae (where the bias is almost complete towards only one of the six codons for Leucine), whereas in Dacinae, a lower bias is observed. A similar pattern is observed in the TTT (Phe) and AAA (Lys) codons (see Figure 2 and Appendix A).

This synonymous preference for codons with adenine and thymine in the third position explains the elevated A + T content in mitogenomes. This feature has been previously observed in Dacinae, specifically in *B. zonata* and *B. minax* [35,53], and other members of Tephritidae, such as *Sphaeniscus atilius* [64]. In other Diptera, the same trait has been reported in Calliphoridae (*Cochliomyia hominivorax*) [69], Ulidiidae (*Tetanops sintenisi*) [70], and species of Tachinidae, Culicidae, Oestridae, and Sarcophagidae [65], as well as in other Insecta, in Neuroptera [60], Hymenoptera, and Aphelinidae [71]. This A + T skew may be the result of selection for translational efficiency or a bias in mutational tendency in mitochondrial genomes [72]. Previous studies have postulated that the possible origin of synonymous codon usage bias (SCUB) could be associated with mutational bias alone or with both mutation bias and natural selection (purifying) [73].

We obtained an average CAI value of 0.46 ± 0.0337 for Tephritidae, with the highest values found in the Dacinae subfamily, specifically for the *nad4*, *cob*, and *nad1* protein-coding genes. The *cob* gene with a peak of 0.515 was the most expressed, followed by *nad5* with a peak of 0.508. The lowest value for Dacinae was for *nad4l* with a peak of 0.456, being the least expressed gene. Similar values of gene expression have been reported in other taxonomic groups, such as Reptilia, with 0.79 for *cob* [74], and a high overall coding region index ranging from 0.40 to 0.78 in different *Amanita* species [75], which could be related to high translational efficiency in mitochondria [30].

The single species of Phytalmiinae represented in this study showed high CAI values but no significant differences among genes, similar to the pattern in Tephritinae. This bias presented different patterns by gene when compared within Tephritinae and the single species in Phytalmiinae, although values are not significantly different as in the Dacinae and Trypetinae, resulting in a complete bias in the entire coding region (Figure 3 and Appendix A). Comparable CAI values have been noted across various species, including *Drosophila*, ants, honeybees, *Nasonia*, *Anopheles* mosquitoes, and Hymenoptera. This indicates that this feature is prevalent among insects, despite minor differences among taxonomic groups [76,77]. Likewise, in Lepidoptera, particularly *Bombyx mori*, previous analyses have shown that mitochondrial DNA may undergo different selective pressures in protein-coding genes [30]. CAI measures the relative efficiency of codon usage in a gene compared with codon usage in highly expressed genes (with values > 0.5) [78]. Therefore, the values we obtained for these genes among the studied species might reflect potential differences in gene expression, which can be also a product of adaptive evolution driven by ecological pressures, energy demands and the need for efficient protein synthesis, specifically at this mitochondrial level, where function is tightly linked to survival and fitness of species [30,73].

Metabolically, the NADH genes (subunits 1, 4, and 5) and the *cob* gene, which belong to complexes 1 and 3, respectively, of the electron transport chain, play important roles within mitochondria in essential processes such as oxidative phosphorylation and energy production [79,80,81]. Insects have a high metabolic rate, which has been analyzed in different organisms [46,82,83]. Various species also have adaptations to temperature, altitude, availability of resources, and other attributes [41,84].

Our analysis of nucleotide composition asymmetry and CAI in *cob* and the NADH genes confirms a differential pattern among taxonomic groups, where we found genetic variability in some species within subfamilies. In Dacinae, there are generalist and specialist species in host plant usage [85,86,87]. The generalists, which are more likely to be of economic importance, have generally been more studied, are easier to obtain for molecular analysis [85], and thus are more heavily represented in our analyses.

In Trypetinae, specifically in some species of *Rhagoletis*, where most have been reported as specialists [88], we found that the genetic variability in mitochondrial genes could provide insights into understanding species radiation and host use evolution within the family. However, a more in-depth analysis is required, incorporating more species that conform to subfamilies, to attempt to elucidate an evolutionary behavior within the family.

In addition, codon adaptation index values could infer the effect of selection in generating this pattern observed in these genes, especially in the frequency of codon usage [89], which makes it useful to understand selection processes in insect mitogenomes. In this sense, the particular pattern of bias observed in a given species is thought to be the product of drift and selection pressures acting on a number of parameters but mainly on tRNA gene copy number and genomic %GC content [90,91].

Our results show an association between codon bias in the coding region with high variation in nucleotide composition in tRNA genes, which would support this previous finding. However, future molecular evolutionary analyses of mitochondrial tRNA genes may provide further evidence in favor of this hypothesis.

### 2.4. Ka/Ks Ratio and Neutrality Plot Analysis

In order to confirm the phylogenetic patterns associated with molecular variability in the mitochondrial genome, we analyzed the gene evolutionary rate in Tephritidae by using neutrality analysis for all PCGs. We found high evolutionary rates in *nad4l*, *nad4*, *nad5*, and *nad1. nad4l* and *nad4* obtained >1 values (an average of 1.5 and 1.2 approximately), and *nad5* had an average of 1 and *nad1* an average of 0.9. We observed COX genes (subunits 1, 2, and 3), and *cob* and *atp6* genes with the lowest evolutionary rates (Figure 4A and Appendix A).

Low evolutionary rates in COX genes have also been observed in different Tephritidae species, such as *B. zonata* and *D. vijaysegarani* [35,37]; in other Diptera, including species of Syrphidae and Psilidae [92,93]; and in Hemiptera, specifically the Heteroptera suborder and the Pentatomomorpha infraorder [94]. This could suggest the use of these genes, especially *cox1*, as genetic markers in Tephritidae [35,47,57,95], due to their being under purifying selection, which results in preserved function and genomic sequence conservation across large evolutionary time scales of the gene [96,97].

Genes with high evolutionary rates were evaluated within subfamilies (Figure 4B and Appendix A), showing differential results due to the number of species in each subfamily; however, the analysis of the Ka/Ks values in the NADH genes varied across the subfamilies. In Dacinae and Trypetinae, we observed similarities caused by their high number of species, demonstrating an appreciable distribution and therefore more variability explained in a higher number of synonymous substitutions (Figure 4B and Appendix A). This evidence proves that Dacinae present a differential behavior, with atypical values (see Figure 1C) that continue to evidence a strong bias of this subfamily, as we have found in a previous analysis. We observed high substitution rates in individual genes (*nad4* and *nad4l*) and a high bias obtained in the CAI, demonstrating high expression levels, which is consistent with the metabolic requirements of organisms and lifestyles; this also suggests the presence of neutral species and species under selection processes.

To corroborate previous results, (skew analysis, CAI, and Ka/Ks), we performed a neutrality plot analysis of the genes with high evolutionary rates; this allowed us to demonstrate important differences in 4 out of 13 PCGs in the mitogenomes of some members of Tephritidae. We found natural selection as the main influence (values ranging from 80 to 86% approximately) and mutational pressure effects in a low percentage (values ranging from 15 to 22% approximately) (Figure 5 and Appendix A), Likewise, we found high R^2^ values, which could be correlated with bias in the third-codon position obtained in the CAI or RSCU results, confirming a strong variation within some members of the family, even at the individual genetic level. Also, our results are congruent with what has been analyzed in other members of Diptera and Lepidoptera, such as *Blepharipa* sp. and *Bombyx mori*, respectively, concluding that natural selection plays an important role in the codon usage of these mitochondrial genes [65].

Neutrality analysis has been useful to elucidate mitochondrial genome evolution as in *Bactrocera* and *Dacus* [35,37,41,53]; however, Yong et al. [37] and Zhang et al. [53] discussed genetic variability, but its significant contribution to cellular energy metabolism remains unclear. Our analysis aims to address this by confirming that specific codon biases, which lead to high expression levels and evolutionary rates, are associated with different genetic plasticity in some Tephritidae species. For instance, some Dacinae members, being generalists, can colonize multiple fruit hosts and become economically parasitic. This could explain similar behaviors in other subfamilies.

Extrinsic factors such as distribution, temperature, and resource availability have been studied to determine mitochondrial evolution in *Bactrocera* [41]. Mitochondrial genes involved in oxidative phosphorylation and energy production, such as the NADH genes (subunits 1, 4, 4L, and 5), have shown positive selection, indicating their significant role in adaptation [41]. This adaptation could allow species to exploit various hosts and survive environmental changes, contributing to the generalist adaptation seen in some Tephritidae members. Another case could be explained by adaptations to specific host plants in fruit flies that are linked to genes under selection. Neutrality tests have revealed deviations in genes related to olfactory receptors, emphasizing their importance in detecting and adapting to host plants [86]. The selection of these genes likely allows fruit flies to specialize in using a limited range of host plants, a crucial trait for their survival and reproduction [86].

Temperature variability impacts insect metabolism, as evidenced in Lepidoptera moths, some Coleoptera, Orthoptera, and phytophagous species. Increased temperatures elevate metabolic rates, demanding higher energy in all life stages [98]. Insects exposed to seasonal temperature changes may exhibit greater plasticity compared with those in constant tropical climates, possibly requiring more host availability.

To validate our hypothesis on adaptive evolution in Tephritidae, a comprehensive approach integrating more members of the family and nuclear genome data is necessary for a deeper understanding of their evolutionary dynamics.

### 2.5. Phylogenetic Reconstruction of Tephritidae

We obtained a fully supported tree, with congruent bootstrap values (>50), that clearly defined the presence of four subfamilies within the family. The substitution model used for our data according to ModelFinder was GTR + F + R5, allowing us to find subfamilies and tribe groupings (Figure 6). Different approaches have been used to attempt to resolve phylogenetic relationships within the family [9,36,48,99,100,101], in particular to solve relationships among *Bactrocera*, *Dacus*, and *Zeugodacus*. According to our results, *Bactrocera* and *Dacus* are more closely related with the majority of *Zeugodacus* as sister groups to that clade. *Zeugodacus* is paraphyletic with *Z. cilifer* as a sister group to all other Dacini (Figure 6). However, due to other phylogenies, these relations continue to be a debate for the family that should continue to be analyzed, specifically to determine if *Zeugodacus* can behave as an independent genus [36,99].

At the tribe level, our results are consistent with previous studies, with Dacini and Ceratitidini as sister groups belonging to Dacinae, separate from Toxotrypanini (Trypetinae subfamily), represented here by *Anastrepha* [9], (Figure 6), which is different from what has been reported on Ceratitidini and Toxotrypanini tribes, i.e., that they are more related compared with Dacini [36,99,102]. This leaves open the debate on the phylogenetic relationships of “higher Tephritidae” [8], which have been possible to analyze and discuss due to mitochondrial genes and the important information in protein-coding genes.

We considered the phylogeny produced to investigate the evolution of lifestyle characters and species distribution within some members of the family. We found that ancestral clades such as Tephritinae members (according to our phylogeny) are specialists adapted, which was maintained in most members of the Trypetinae subfamily (Carpomyini tribe), except for *Anastrepha fraterculus*, only member of the Toxotrypanini tribe, which can be linked to the ‘*fraterculus* complex’, being more generalists (Figure 6). In Dacinae, we found both adaptations highlighting the Ceratitidini tribe as generalists, as previously evidenced [103]; for Dacini, we emphasize members of *Zeugodacus* as specialists, using mostly fruits from the Cucurbitaceae family (Appendix A), while for species such as *Dacus trimacula*, there is a preference for hosts not only from the Cucurbitaceae family but also from some Asclepediaceae and Apocynaceae, especially flowers and fruits (Appendix A).

Finally, *Bactrocera* species have been reported as generalists [104,105,106], using multiple hosts but preferring families such as Myrtaceae, Rosaceae, Anacardiaceae, and Fabaceae, among other economic and commercial families (Appendix A). Most Tephritidae individuals are distributed in Asia, but they have been introduced into new continents, including Australia, as in the case of *Procecidochares utilis*, or present restricted distribution ranges for certain regions, like some members of *Ceratitis*, *Merzomyia westermanni*, and *Tephritis californica*, along with others [107] (Appendix A).

According to our results, we found evidence that could explain how the adaptation of generalist species (such as some members of Dacinae and Trypetinae) may facilitate evolution within Tephritidae. The genetic variability observed in these organisms, specifically in the NADH genes, evaluated through analyses of asymmetry, codon bias, CAI, substitution rate, and neutrality values, could influence their ability to exploit fruit hosts, as in the case of some members of *Bactrocera* [41]. This suggests a possible relationship between their adaptations and extrinsic factors such as resource availability and competition, which may contribute to the diversity of lifestyles in these organisms. Thus, the preference of generalist organisms for multiple hosts could be the key to the evolution of Tephritidae populations.

Accordingly, some members of Tephritinae, such as *Procecidochares utilis* and species of *Tephritis*, exhibit a specialist feature, exclusively associating with specific parts of plants from the Asteraceae family (flowers) [2,7]. This specialist feature could be related to low molecular variability, reflected in low asymmetry, bias, nucleotide composition, and CAI, as well as clear differentiation in the nucleotide substitution rate, observed in our results, but more species should be taken into consideration to reflect this.

Finally, climate change and global warming are impacting the expansion, reproduction, and survival of these organisms [108]. These environmental pressures could be shaping adaptations and their genetic bases, indicating new directions in the evolution of these insects and highlighting their economic importance worldwide.

## 3. Methods

### 3.1. Sampling of Genomic Data in Tephritidae

We searched the Sequence Read Archive (SRA) NCBI database (https://www.ncbi.nlm.nih.gov/sra, accessed on 15 May 2024) and downloaded genomic or transcriptomic data for Tephritidae. We included genomic or transcriptomic data from some Tephritidae genera in which either no mitochondrial genome sequences have been reported or only a few sequences have been recorded in databases. We built an information table with data characteristics such as taxonomic classification, sequencing method, assembled reads, and others (Appendix A).

### 3.2. mtDNA Assembly and Extracted Read Assembly

Initially, we checked read quality by using FastQC v0.12.1 software [109], obtaining a >30 Phred value by sequence; then, the totality of reads from each species was used to assemble the mitochondrial genomes by using MITGARD v1.0 [110] with default parameters, using PAIRED-END mode and a single reference in FASTA format. MITGARD maps all reads to a reference mitogenome by using the program Bowtie2 v2.5.4 [111] and then uses Trinity v2.15.1 software [112] and Spades v4.0.0 [113] to assemble the mitochondrial genome from the previously mapped reads; finally, it performs a second mapping of the assembled contigs to the reference mitogenome with the program Minimap2 v2.28 [114], using default parameters to run all algorithms to obtain a final mitochondrial genome. In the same way, we used Canu assembler [115] for long-read data obtained from long-read sequencing (PacBio or Nanopore), with default parameters, to assemble PacBio HiFi data; subsequently, we obtained a single value of assembled reads for each sequence with Samtools v1.21 software [116]. For each organism, we performed in parallel a second assembly by mapping with the Bowtie2 tool in Geneious Prime 2020.0.5 [117] based on a low-sensitive strategy to fill gaps and missing nucleotide strings in PCGs. From these two mitogenomes assembled independently for each species (MITGARD v1.0 + Geneious software v2024.0), a consensus sequence was obtained in order to reduce possible gaps, eliminate ambiguous bases and reduce the possibility of false rearrangements (e.g., false inversions in tRNAs), obtaining high-quality and complete mitochondrial genomes from raw reads in this study. We used multiple reference mitogenomes with both strategies: *Bactrocera dorsalis* (GenBank accession number NC_008748), *Dacus bivittatus* (NC_046468), *Rhagoletis cerasi* (NC_061399), and *Tephritis femoralis* (NC_047184). Only mitogenomes obtained with a total coverage of 80% (>1000× average on depth of coverage) and protein-coding genes (PCGs) coverage of 95% or more were taken into account for the analysis. Information on the genomic data used and the process of mtDNA assembly are summarized in Appendix A.

### 3.3. mtDNA Sequence Manual Annotation

We used a semi-automatic gene annotation methodology revising the presence and correct disposition of protein-coding genes (PCGs), transfer RNA genes (tRNAs), and ribosomal RNA genes (rRNAs) by (i) using a combination of software such as MITOS2 web server (http://mitos2.bioinf.uni-leipzig.de/index.py, accessed on 2 June 2024) [118] and tRNAScan (https://lowelab.ucsc.edu/tRNAscan-SE/, accessed on 5 June 2024) [119] with the invertebrate genetic code and (ii) performing the manual confirmation of the genetic annotation by using Geneious Prime 2020.0.5 with clustal Omega alignment [120] to corroborate gene coordinates and start/stop codons in each PCG.

### 3.4. Sampling and Mitogenome Sequence Review

We retrieved a total of 44 complete mitochondrial genome sequences available for Tephritidae, as of October 2022, from the Organelle—RefSeq database NCBI (https://www.ncbi.nlm.nih.gov/genome/browse#!/organelles/, accessed on 15 October 2022). These and the ten species for which new mitogenomes were obtained in this study are listed in Appendix A. Each sequence was classified by its tribe and subfamily according to the NCBI Taxonomy Browser and Integrated Taxonomic Information System (ITIS). A numerical gene order was made (from 1 to 37, with *trnI* as 1, *trnQ* as 2, etc.) and the gene orientation indicated according to the position of the gene in the heavy strand (+) or light strand (−). For this analysis, the control region (CR) was not considered due to the absence of sequencing or annotation (complete or partial) in a significant number of mitochondrial genomes. The ancestral gene order of the insect mitochondrial genome postulated by Cameron [23] was used to identify possible reorganizations (duplication, deletion, and inversion–translocation) by the observational analysis of the genes and their annotations by using Geneious Prime 2020.0.5. The order and orientation of the genes in each mitogenome analyzed is summarized in Appendix A.

### 3.5. AT/GC Skew in Mitochondrial Genomes

The nucleotide composition values from Tephritidae mitogenomes obtained in this study and downloaded from the NCBI RefSeq database resulted from four data sets: (i) whole-mitochondrial-genome sequences; (ii) 13 extracted and concatenated protein-coding genes (PCGs; gene order in plus/plus orientation was *atp6*, *atp8*, *cox1*, *cox2*, *cox3*, *cob*, *nad1*, *nad2*, *nad3*, *nad4*, *nad4L*, *nad5*, *and nad6*); (iii) two rRNAs (*rrnL* and *rrnS*); and (iv) 22 tRNAs genes (gene order in plus/plus orientation was *trnI*, *trnQ*, *trnM*, and so on). We identified strand asymmetry in the nucleotide composition (GC and AT skewness) of each data set by applying the formulae GC skew = (G − C)/(G + C) and AT skew = (A − T)/(A + T) by using the software SeqTK: Toolkit for processing sequences in FASTA/Q formats (https://github.com/lh3/seqtk, accessed on 3 August 2024). Finally, we used a scatter plot obtained in ggplot2 [121] in RStudio v9.4.191303 software [122] to find relations among the genes.

### 3.6. Relative Synonymous Codon Usage (RSCU) and Codon Adaptation Index (CAI)

We calculated the CAI and RSCU for each of the 13 concatenated PCGs in the data set (not including stop codons) by using CAICal v1.0 software [123]. We calculated RSCU values for the 60 sense codons to investigate for potential codon usage bias (CUB) on Tephritidae mitogenomes and plotted heatmaps of these data by using pheatmap [124] in RStudio v9.4.191303.

We calculated the codon adaptation index (CAI) for each of the 13 PCGs of each Tephritidae species, using as reference the codon usage table for *Drosophila melanogaster*, available in the codon usage database (http://www.kazusa.or.jp/codon/, accessed on 15 August 2024). To calculate these values, we used a sliding window methodology with windows of 300 bp (i.e., 100 codons), processed through Perl scripts (https://github.com/CaioFreire/CUB, accessed on 20 August 2024); finally, the results were analyzed numerically to estimate which genes are highly expressed within each subfamily of Tephritidae. The results were plotted by using ggplot2 [121].

### 3.7. Ka/Ks Ratio and Neutrality Plot Analysis

We performed substitution tests to estimate ratios among neutral, beneficial mutations and purifying selection acting in protein-coding genes by using a multifasta file for each PCG of every Tephritidae species with the PCGs of *Drosophila melanogaster* as reference. We aligned these data with Clustal Omega and then analyzed them in DNASp6 v6 software [125] and produced boxplot graphs at the family and subfamily levels with ggplot2 [121] in RStudio.

Neutrality plot analysis was used to estimate how extrinsic factors influence the preference of a certain use of codon towards mutational pressure and natural selection. To obtain this, we retained genes with high Ka/Ks values (1 and >1) for more detailed analyses. We used CodonW v1.0 software [126] to obtain information about the percentages of GC usage in codon positions 1, 2, and 3. Finally, through scatter plots created with ggplot2 [121] in RStudio, we reviewed the interaction between the average of GC usage in positions 1 and 2 against GC usage in position 3. The slope value in the equation that explains the analysis determined which factor influenced the conduct of the chosen PCGs.

### 3.8. Phylogenetic Inference, Lifestyle Adaptations, and Species Distribution in Tephritidae

We performed phylogenetic inference based on the 54 Tephritidae taxa here studied, the 10 newly sequenced species here produced, and 44 previous published mitogenome Tephritidae sequences (Appendix A), with the *Drosophila melanogaster* mitogenome (NC_024511) as the outgroup. In Geneious, we extracted and concatenated all 13 PCGs and 2 rRNAs genes of each mitogenome. The multifasta file with nucleotide information was then aligned in Clustal Omega web server [120], and Gblocks web server [127] was used to remove low-identity regions (less than 60% of identity) from the alignment. Finally, by using the IQ-TREE web server, we obtained a maximum likelihood phylogenetic tree, with bootstrap support of 1000 replicates, and used ModelFinder v1.4.2 [128] to find the best substitution model for the nucleotide data. We then built a distribution and lifestyle matrix for each species of the family (Appendix A) with data obtained from the Global Database EPPO (https://gd.eppo.int/, accessed on 15 July 2024) and digital library CABI (https://www.cabidigitallibrary.org/, accessed on 15 July 2024), with the purpose of finding relationships among the selection processes of the genes, different adaptations, and different lifestyles related with phylogenetic features to explore the evolution of Tephritidae.

## 4. Conclusions

According to the data obtained and analyzed from the mitochondrial coding region of some members of Tephritidae, we found evidence of associations in AT/GC skew analysis, codon adaptation index, and substitution and neutrality analyses. Our analysis shows that some members of Tephritinae presented less asymmetry and bias in nucleotide composition as common molecular feature associated with relaxed selection, opposite to what we obtained in some members of Dacinae and Trypetinae. Similar results were found by analyzing bias on single genes of the coding region, focusing on genes with the highest expression values that, at the same time, presented metabolic importance in the NADH complex and could play important roles in the adaptation process of species.

Our phylogeny also allowed us to contribute to the debate on phylogenetic relationships and helped us to demonstrate possible patterns among molecular variability, adaptations, and lifestyles, associated with host specificity in some members of Tephritidae, allowing us to consider generalism as a possible predominant adaptation in some members of the family, resulting in a preference of organisms for multiple hosts and possibly being the key to the evolution of Tephritidae populations; however, to confirm this, future investigations should consider similar analysis with more species of the family in nuclear genes and deeply investigate the mitochondrial non-coding region. In our case, we found a strong phylogenetic signal especially in the AT/GC skew values of tRNAs; however, we did not focus on these genes due to the lack of evolutionary information offered to us, compared with the coding region; this is mainly due to a low number of direct studies focused on evolution, where we could show variation in non-coding regions, but analysis and investigations in this region would be interesting to understand the adaptative process and continue to obtain evidence for Tephritidae evolution.

Finally, an evolutionary analysis could be implemented in pest control and management strategies by addressing potential weaknesses in adaptation to certain hosts [129]. Pest control could exploit these vulnerabilities by disrupting the ecological niches necessary for their survival [129]. Additionally, species-specific strategies can be implemented, incorporating biological control and predicting the development of resistance to pesticides through the analysis of specific genes involved in fruit recognition [130].

## Figures and Tables

**Figure 1 ijms-26-05560-f001:**
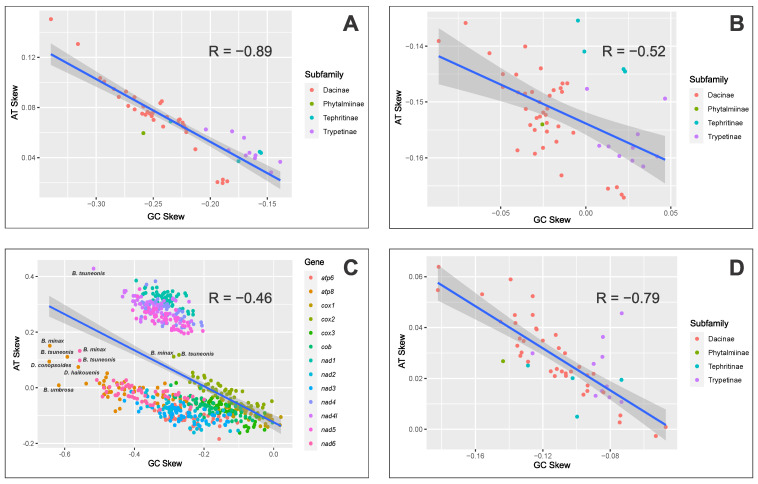
Tephritidae AT/GC skew. (**A**) Whole-genome AT/GC skew per subfamily with inversely proportional relation and Pearson coefficient of −0.89. (**B**) Protein-coding gene AT/GC skew per subfamily with inversely proportional relation and Pearson coefficient of −0.52. (**C**) Individual protein-coding gene AT/GC skew presenting outliers from Bactrocera and Dacus genera, inversely proportional relation, and Pearson coefficient of −0.46. (**D**) Transfer RNA gene AT/GC skew per subfamily with inversely proportional relation and Pearson coefficient of −0.79. Axis levels are normalized AT/GC ratios obtained by applying analysis formulae.

**Figure 2 ijms-26-05560-f002:**
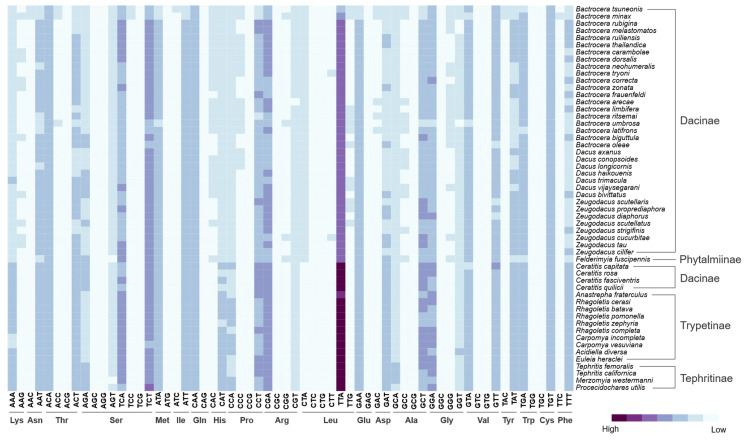
Heatmap of relative synonym codon usage of protein-coding genes. Purple indicates higher usage, and light blue indicates lower usage of codons according to the encoded amino acid.

**Figure 3 ijms-26-05560-f003:**
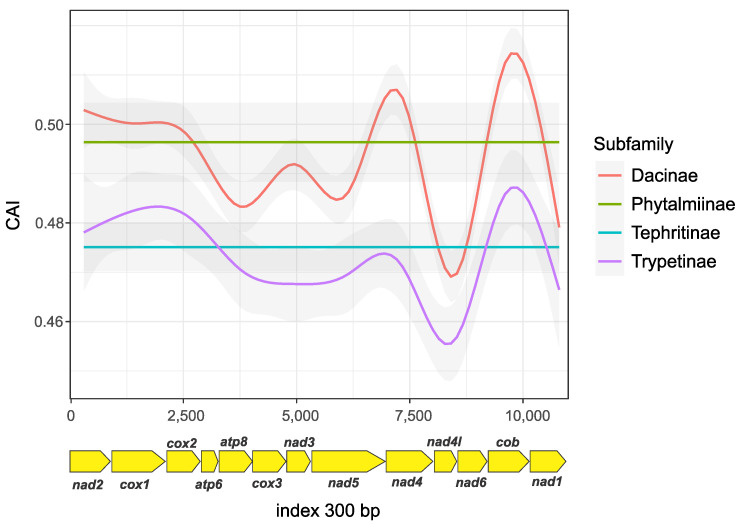
Tephritidae codon adaptation index (CAI). Index of protein-coding genes by subfamily (each represented by a colored line) calculated through windows of 300 pb.

**Figure 4 ijms-26-05560-f004:**
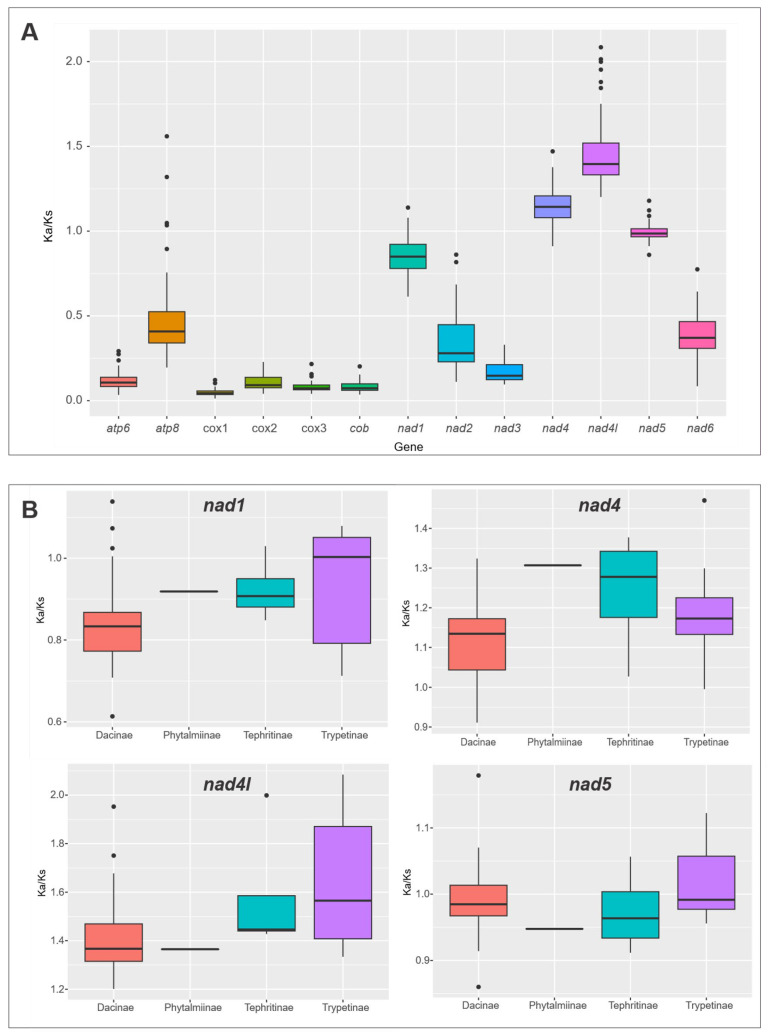
Ka/Ks ratio Tephritidae. (**A**) Boxplot of Ka/Ks evolutionary rates by protein-coding genes and (**B**) boxplot of Ka/Ks ratios of genes with high evolutionary rates within subfamilies. Genes or subfamilies with ratios higher than 1 are under positive selection, and those with ratios lower than 1 are under purifying selection. Those with a ratio of exactly 1 are under neutral selection.

**Figure 5 ijms-26-05560-f005:**
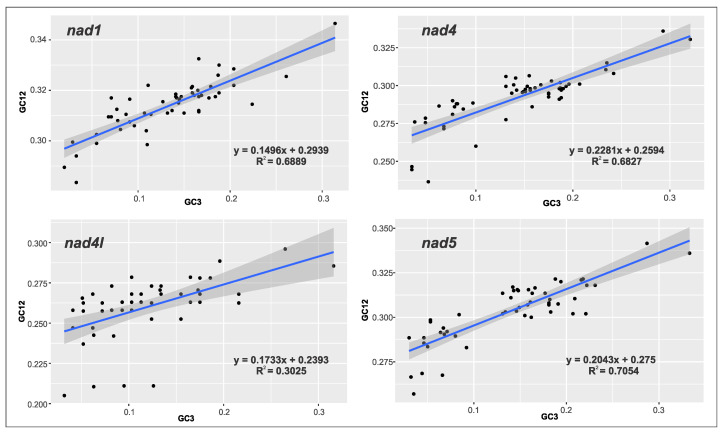
Neutrality plot of genes with high evolutionary rates. Levels of natural selection are higher than mutational pressure according to the slope in each graph.

**Figure 6 ijms-26-05560-f006:**
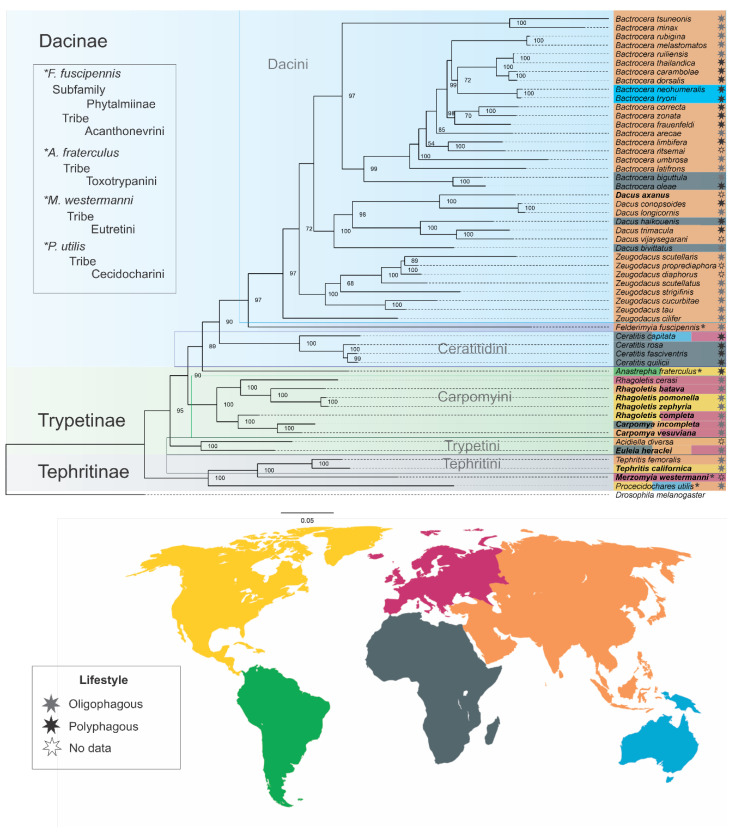
Tephritidae phylogenetic reconstruction. The tree was obtained with the maximum likelihood analysis of concatenated and the alignment of 13 protein-coding and 2 ribosomal genes from the mitochondrial genome. A total of 54 tephritid species were included, and *Drosophila melanogaster* was included as an outgroup. The values at the top of each node represent strong bootstrap support. The colors highlighting the name of every species match the distribution according to the map, and the stars represent the lifestyles. Asterisk (*) indicates that the analyzed species are unique to the subfamily or tribe to which they belong.

## Data Availability

The original contributions presented in this study are included in the article/Appendix A. Further inquiries can be directed to the corresponding author.

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
