# Peer review of "Mitochondrial Genome Variations and Possible Adaptive Implications in Some Tephritid Flies (Diptera, Tephritidae)"

_ijms, 2025, doi:10.3390/ijms26125560_

Round 1
Reviewer 1 Report
Comments and Suggestions for Authors
I have read a lot of the authors' past publications and based on that, I read the current manuscript and found it to be quite reasonable and suitable for publication.
The authors have done a deep study on evolution through the analysis of the mitochondrial genome. The authors' analysis method, results, and conclusions are reasonable and I think it will be an excellent reference for other scientists in the future.
I also do peer review for quite a few journals.
Most of the manuscripts submitted to these journals are about mitochondrial genome sequencing, assembly, and phylogenetic tree drawing, and the conclusion that this will help in evolutionary research.
Most of them are not very original. However, this manuscript that IJMS asked me to review is very original compared to other papers and provides biogeographic insights.
The supporting analysis data is also more abundant than other papers.
Therefore, I have agreed to publish it in its current form.
Author Response
Comments 1: I have read a lot of the authors; past publications and based on that, I read the current manuscript and found it to be quite reasonable and suitable for publication.
The authors have done a deep study on evolution through the analysis of the mitochondrial genome. The authors; analysis method, results, and conclusions are reasonable and I think it will be an excellent reference for other scientists in the future.
I also do peer review for quite a few journals.
Most of the manuscripts submitted to these journals are about mitochondrial genome sequencing, assembly, and phylogenetic tree drawing, and the conclusion that this will help in evolutionary research. Most of them are not very original. However, this manuscript that IJMS asked me to review is very original compared to other papers and provides biogeographic insights. The supporting analysis data is also more abundant than other papers. Therefore, I have agreed to publish it in its current form.
Response 1: Thank you for pointing this out. We thank the reviewer for his comments on our study.
Reviewer 2 Report
Comments and Suggestions for Authors
The manuscript presents a very interesting approach to studying mitochondrial sequences in a potentially relevant species. In my opinion, the manuscript could be published after minor adjustments.
- Lines 11-26: The abstract mentions "10 new mitochondrial genomes" and "selection pressures in certain NADH genes." However, it does not clearly summarize key methods used (e.g., bioinformatics tools) or the implications of these findings in broader evolutionary biology. Suggest adding brief mentions of tools and the practical significance.
- Line 36: "may be related to molecular evolution in mitogenomes" — Remove "may be" for a more definitive tone.
- Line 43: Typo in "constituting a serious food security problem around the world" — consider rephrasing to "posing significant food security challenges worldwide."
- Line 85: "Complete mitochondrial genomes" — Consider rephrasing to "Comprehensive mitochondrial genome datasets."
- Line 112: "Assembled mitogenomes and features" — consider clarifying why these 10 species were chosen (e.g., ecological or evolutionary relevance). Typo in "mitochondrial reads" — ensure clarity in referring to read depth or sequence length.
- Lines 135-146: The nucleotide composition analysis is thorough but lacks statistical tests to confirm significant differences between subfamilies.
- Lines 222-239 (Figure 2): The discussion of codon bias would benefit from elaboration on how these patterns impact translational efficiency or adaptive evolution.
- Selection Analysis (Lines 304-314): The neutrality analysis mentions evolutionary rates but does not clearly link these to ecological or behavioral adaptations. Provide additional examples or case studies for context.
- Line 307 (Figure 4a): The use of Ka/Ks ratios to infer selection is valid but could be strengthened by comparing these results to nuclear genes or other taxa.
- Lines 373-424: The phylogenetic analysis results are presented well, but the discussion on lifestyle adaptations is speculative. Provide more concrete examples or references supporting these claims.
- Lines 437-549: The description of bioinformatic tools is detailed but lacks information on parameter settings for MITGARD, Bowtie2, and other software. Specify these for reproducibility.
- Figure 1 (Line 165): Axis labels are missing units or descriptive text, which could help in interpreting the skew values.
- Table 1 (Line 147): The table could include a column summarizing major findings or insights for easier interpretation.
- Line 552 (Conclusion): While the conclusion is clear, it could emphasize the implications for pest control or conservation of Tephritidae flies.
Author Response
|
Comments 1: Lines 11-26: The abstract mentions "10 new mitochondrial genomes" and "selection pressures in certain NADH genes." However, it does not clearly summarize key methods used (e.g., bioinformatics tools) or the implications of these findings in broader evolutionary biology. Suggest adding brief mentions of tools and the practical significance. Response 1: Thank you for pointing this out. We include the main bioinformatics tools used in the analyses (see lines page 1, lines 17, 20, 22, 23), as well as the practical significance of the study (see lines 26 to 30).
|
|
Comments 2: Line 36: "may be related to molecular evolution in mitogenomes" — Remove "may be" for a more definitive tone. |
|
Response 2: Agree. Expression changed in the manuscript, page 1, line 41.
Comments 3: Line 43: Typo in "constituting a serious food security problem around the world" — consider rephrasing to "posing significant food security challenges worldwide." Response 3: Agree. Typo corrected in the manuscript, page 2, line 50.
Comments 4: Line 85: "Complete mitochondrial genomes" — Consider rephrasing to "Comprehensive mitochondrial genome datasets." Response 4: Agree. Corrected in the manuscript, page 2, line 92.
Comments 5: Line 112: "Assembled mitogenomes and features" — consider clarifying why these 10 species were chosen (e.g., ecological or evolutionary relevance). Typo in "mitochondrial reads" — ensure clarity in referring to read depth or sequence length. Response 5: It was clarified in the manuscript why these 10 species were chosen, specifically in page 3, lines 127-128. According to "mitochondrial reads," we refer to the fact that the assembly resulted in reads that belonged to the mitochondria of these organisms. In this case, the reads ranged from 20,113 bp to 182,047 bp. "In length" is added for greater clarity in line 131 page 3.
Comments 6: Lines 135-146: The nucleotide composition analysis is thorough but lacks statistical tests to confirm significant differences between subfamilies. Response 6: Thank you for pointing this out. In the analysis of nucleotide composition, a correlation between AT skew and CG skew was performed. This correlation analysis between these two parameters is classically performed to compare whether there is asymmetry in the nucleotide composition in the mitogenome between the subfamilies, which is evidenced in Figure 1. In addition, the low number or large differences between the mitogenomes analyzed within each subfamily makes statistical analysis to determine significant differences of mitogenomes of limited utility.
Comments 7: Lines 222-239 (Figure 2): The discussion of codon bias would benefit from elaboration on how these patterns impact translational efficiency or adaptive evolution. Response 7: Agree. The discussion on this issue was broadened and a new reference supporting this has been added specifically page 10, lines 284 to 287.
Comments 8: Selection Analysis (Lines 304-314): The neutrality analysis mentions evolutionary rates but does not clearly link these to ecological or behavioral adaptations. Provide additional examples or case studies for context. Response 8: Agree. The discussion on this issue was broadened and a new reference supporting this has been added. See page 14, lines 379 to 383.
Comments 9: Line 307 (Figure 4a): The use of Ka/Ks ratios to infer selection is valid but could be strengthened by comparing these results to nuclear genes or other taxa. Response 9: Strongly agree. Actually, our conclusion is to obtain more results using more taxa or members of the family and contrast results with nuclear genes. As investigation group, we are beginning new analysis in chemoreceptors genes such as odorant families. However, these analyses are being performed on the basis of new whole genome sequences and are not yet included in this study.
Comments 10: Lines 373-424: The phylogenetic analysis results are presented well, but the discussion on lifestyle adaptations is speculative. Provide more concrete examples or references supporting these claims. Response 10: This is the novelty of our results, as we have not found a lot of investigations focused on the understanding of mitochondrial genes variation and adaptation to lifestyles. In line 440 of page 16, we cited a key reference to our work, which tried to evaluate these variations but at genera level, in this case, members of Bactrocera.
Comments 11: Lines 437-549: The description of bioinformatic tools is detailed but lacks information on parameter settings for MITGARD, Bowtie2, and other software. Specify these for reproducibility. Response 11: Agree. Parameters of bioinformatic tools were added specifically in page 16, lines 468 to 469 and page 17, lines 473 to 476.
Comments 12: Figure 1 (Line 165): Axis labels are missing units or descriptive text, which could help in interpreting the skew values. Response 12: Agree. Information added in page 7, lines 182 – 183.
Comments 13: Table 1 (Line 147): The table could include a column summarizing major findings or insights for easier interpretation. Response 13: Agree, in fact one of the reviewers suggest converting this table in a Supplementary Table, so it is going to be present as a supplementary material (see also response to comment 3 of reviewer 3).
Comments 14: Line 552 (Conclusion): While the conclusion is clear, it could emphasize the implications for pest control or conservation of Tephritidae flies. Response 14: Agree. The discussion on this point was expanded and also references has been added to complement those implications specifically in page 19, lines 596 to 601.
Reviewer 3 Comments 1: The authors appear to overstate the adaptive significance of mitochondrial genome evolution. Given that mitochondrial genomes have long been considered to evolve under neutral or nearly neutral conditions, the probability of positive selection being maintained is relatively low. The authors' claim about positive selection on NADH genes (Line 19) contributing to adaptive evolution in tephritid flies seems overstated. I suggest using more conservative and rigorous language when discussing these findings Response 1: Thank you for pointing this out. the language was changed and generated a new sentence (see lines 26 to 30).
Comments 2: The authors repeatedly discuss patterns at higher taxonomic levels (subfamilies), which is too broad for meaningful discussion of species-specific adaptations. The research objectives and content need to be more clearly defined and aligned. The authors should clearly state whether their focus is on broad phylogenetic patterns or species-level adaptations. Response 2: Agree. We present a clear hypothesis on the role of the molecular variability present in the mitogenomes and its relationship with species-level adaptations in Tephritidae (see lines 114 to 118). In order to answer this hypothesis, we compare the molecular data one by one and between subfamilies to show whether it is a phylogenetic change or possibly associated with adaptive processes.
Comments 3: Table 1 should be moved to supplementary materials as it contains basic information that doesn't substantially contribute to the main conclusions. Response 3: Agree. Table 1 was moved to the supplementary material.
Comments 4: Consider replacing Table 1 with box plots showing GC content trends across different groups. Response 4: Agree. We believe that the information presented in Table 1, once converted into supplementary material, serves its purpose. We consider that Figure 1 better explains the trend in nucleotide content, specifically through the AT/GC skew analysis performed.
Comments 5: For codon usage analysis, I suggest examining amino acid usage frequencies across species. Since codon bias is typically associated with transcription efficiency, demonstrating no species-specific bias in amino acid usage would better establish the role of codon preference in adaptive evolution. Response 5: We indeed want to show that there is a variation of codon usage visible among the Tephritidae subfamilies, tribes and species. The codon usage bias allows us to identify differential transcriptional efficiency between species, which allows us to better infer their role in adaptive processes. In addition, this analysis was complemented with the neutrality test for all coding regions, in order to review the Ka/Ks ratio. Therefore, we do not consider it necessary to include the analysis of amino acid frequencies, since they are implicit in these analyses, particularly in Ka/Ks ratio.
|
Reviewer 3 Report
Comments and Suggestions for Authors
Dear Authors,
I am glad to review your paper. This manuscript presents a study on mitochondrial genome variations and possible adaptive implications in tephritid flies. While the research addresses an important topic in evolutionary biology and genomics, the current version shows several limitations that need to be addressed.
Majors:
1. The authors appear to overstate the adaptive significance of mitochondrial genome evolution. Given that mitochondrial genomes have long been considered to evolve under neutral or nearly neutral conditions, the probability of positive selection being maintained is relatively low. The authors' claim about positive selection on NADH genes (Line 19) contributing to adaptive evolution in tephritid flies seems overstated. I suggest using more conservative and rigorous language when discussing these findings.
2. The authors repeatedly discuss patterns at higher taxonomic levels (subfamilies), which is too broad for meaningful discussion of species-specific adaptations. The research objectives and content need to be more clearly defined and aligned. The authors should clearly state whether their focus is on broad phylogenetic patterns or species-level adaptations.
Minors:
1) Table 1 should be moved to supplementary materials as it contains basic information that doesn't substantially contribute to the main conclusions.
2) Consider replacing Table 1 with box plots showing GC content trends across different groups.
3) For codon usage analysis, I suggest examining amino acid usage frequencies across species. Since codon bias is typically associated with transcription efficiency, demonstrating no species-specific bias in amino acid usage would better establish the role of codon preference in adaptive evolution.
Author Response
Comments 1: The authors appear to overstate the adaptive significance of mitochondrial genome evolution. Given that mitochondrial genomes have long been considered to evolve under neutral or nearly neutral conditions, the probability of positive selection being maintained is relatively low. The authors' claim about positive selection on NADH genes (Line 19) contributing to adaptive evolution in tephritid flies seems overstated. I suggest using more conservative and rigorous language when discussing these findings
Response 1: Thank you for pointing this out. The language was changed and generated a new sentence (see lines 26 to 30).
Comments 2: The authors repeatedly discuss patterns at higher taxonomic levels (subfamilies), which is too broad for meaningful discussion of species-specific adaptations. The research objectives and content need to be more clearly defined and aligned. The authors should clearly state whether their focus is on broad phylogenetic patterns or species-level adaptations.
Response 2: Agree. We present a clear hypothesis on the role of the molecular variability present in the mitogenomes and its relationship with species-level adaptations in Tephritidae (see lines 114 to 118). In order to answer this hypothesis, we compare the molecular data one by one and between subfamilies to show whether it is a phylogenetic change or possibly associated with adaptive processes.
Comments 3: Table 1 should be moved to supplementary materials as it contains basic information that doesn't substantially contribute to the main conclusions.
Response 3: Agree. Table 1 was moved to the supplementary material.
Comments 4: Consider replacing Table 1 with box plots showing GC content trends across different groups.
Response 4: Agree. We believe that the information presented in Table 1, once converted into supplementary material, serves its purpose. We consider that Figure 1 better explains the trend in nucleotide content, specifically through the AT/GC skew analysis performed.
Comments 5: For codon usage analysis, I suggest examining amino acid usage frequencies across species. Since codon bias is typically associated with transcription efficiency, demonstrating no species-specific bias in amino acid usage would better establish the role of codon preference in adaptive evolution.
Response 5: We indeed want to show that there is a variation of codon usage visible among the Tephritidae subfamilies, tribes and species. The codon usage bias allows us to identify differential transcriptional efficiency between species, which allows us to better infer their role in adaptive processes. In addition, this analysis was complemented with the neutrality test for all coding regions, in order to review the Ka/Ks ratio. Therefore, we do not consider it necessary to include the analysis of amino acid frequencies, since they are implicit in these analyses, particularly in Ka/Ks ratio.
Round 2
Reviewer 3 Report
Comments and Suggestions for Authors
I have no further questions. Congrats!